# A Qualitative evaluation in community settings in England exploring the experiences of coaches delivering the NHS Low Calorie Diet programme pilot to ethnically diverse participants

Pooja Dhir [1], Maria Maynard [2,3] Kevin J Drew,[1] Catherine Verity Homer,[4] Chirag Bakhai,[5] Louisa Jane Ells[3]

¹Leeds Beckett University, Leeds, UK
²School of Health, Leeds Beckett University, Leeds, UK
³Obesity Institute, Leeds Beckett University, Leeds, UK
⁴Centre for HEalth and Social Care Research, Sheffield Hallam University, Sheffield, UK
⁵NHS Bedfordshire Luton and Milton Keynes STP, Luton, UK

**Correspondence to**
Pooja Dhir;
p.dhir2675@student.
leedsbeckett.ac.uk

## ABSTRACT

**Background** The management of type 2 diabetes (T2D) within diverse ethnic populations requires a culturally tailored approach. However, little is known about the experiences of coaches delivering interventions for T2D, such as the National Health Service (NHS) Low Calorie Diet (LCD) programme, to people from diverse ethnic backgrounds.

**Objective** To explore the experiences of coaches delivering an NHS programme using total diet replacement approaches to individuals from diverse ethnic backgrounds, to inform the effective tailoring and equitable delivery of future interventions.

**Design** Qualitative study.

**Setting** Individuals delivering the NHS LCD programme.

**Participants** One-to-one semistructured interviews were conducted with seven health coaches delivering the NHS LCD programme. Inclusion criteria included participants delivering the NHS LCD programme either from a minoritised ethnic background or delivering the programme to those from ethnic minority and white British backgrounds.

**Main outcome measures** Qualitative semistructured interviews conducted through different stages of the programme. Reflexive thematic analysis was used to analyse the transcripts.

**Results** Key themes highlighted the following experiences of delivering the LCD programme: (1) training and support needs; (2) needing to understand culture and ethnicity; (3) the impact of language; (4) the use of resources in providing dietary advice and (5) experiences of cultural tailoring. The themes highlight the need to prioritise person-centred care, to integrate culturally tailored approaches and for provision of education and training to those delivering health programmes.

**Conclusion** These findings describe the experiences of health coaches in tailoring delivery and emphasise the role of cultural competence in ensuring equitable and effective healthcare interventions for diverse populations. This learning can inform future programmes and policies aimed at promoting inclusive healthcare practices.

## STRENGTHS AND LIMITATIONS OF THIS STUDY

⇒ The sample did not include representation from all of the service providers delivering the National Health Service (NHS) Low Calorie Diet (LCD) programme as some providers did not engage with the study.
⇒ A potential limitation lies in the small sample size of seven participants; however, it is important to recognise that in qualitative research, the emphasis is placed on the depth and richness of the data rather than on statistical power.
⇒ A strength of this study is its utilisation of one-to-one semistructured interviews, allowing for an in-depth exploration of the experiences of health coaches delivering the NHS LCD programme to individuals from diverse ethnic backgrounds.

concern,[1 2] and its management requires a multifaceted lifestyle approach that encompasses dietary modifications.[1–3] However, food is culturally significant, and it often serves as a cornerstone of traditions, identity and social cohesion.[3] Thus, lifestyle change programmes are advised to consider the impact of culture and ethnicity, among other social factors when attempting to modify the dietary habits of individuals. With an increasingly diverse population in the UK, issues of culture and ethnicity are central to concerns over the equitable provision of healthcare services.[1 4]

Health equity has been defined as the fair and just opportunity, irrespective of social position, to attain full health and well-being from social conditions that seek to promote and support good health.[4] The issue of equity is important because it is healthcare inequities that create, perpetuate and exacerbate health inequalities.[5] Indeed, inequalities in health outcomes and access to healthcare are commonplace and addressing them is a

## INTRODUCTION

The prevalence of type 2 diabetes (T2D) remains a significant global public health

public health priority.[5] For example, in response to the recognition that patients from different ethnic backgrounds may experience disparities in healthcare access and quality,[6] the notion of culturally competent, person-centredness and standardised practices in the delivery of services has been developed to try and meet the need for equitable delivery of healthcare services.[7–9]

Cultural competence in healthcare is described as the ability of healthcare systems to deliver interventions that encompass diversity in cultures, beliefs and behaviours while tailoring the service to individuals' environmental, cultural and social needs.[10] Initially seen as a set of skills and knowledge that healthcare professionals (HCPs) possess to effectively engage with patients from diverse backgrounds,[11] cultural competence has evolved to encompass HCPs' ability to deliver interventions. This delivery should not only accommodate diversity in beliefs and behaviours but also the ability to tailor services to individuals' environmental, cultural and social needs, including different food preferences.[11] HCPs need to possess cultural competences in order to address the needs of the diverse patient populations they deliver to, and thus, deliver healthcare services in a more equitable way.

Research underscores the significance of tailoring health programme advice not only in terms of language but also in culturally appropriate contexts, encompassing written support materials and verbal interactions.[12] Studies emphasise the role of culturally competent communication in healthcare settings, highlighting its positive impact on patient outcomes and satisfaction and indicating that culturally tailored interventions lead to improved health outcomes among diverse populations.[12 13] Therefore, incorporating culturally appropriate language and materials into health programmes is essential for promoting effective engagement and understanding, which can ultimately contribute to better health outcomes and patient experiences.[13] Existing research from the perspectives of patients often underscores the need for culturally relevant support, such as through cultural competency of HCPs and culturally tailored resources such as dietary resources.[14] For example, In the USA, health coaches have been reported to value shared characteristics with their clients which have helped to build a trusting relationship and support with decision-making.[15] More generally, previous research has explored the experiences of HCP–participant interactions and highlighted the significance of coach–participant relationships in enhancing health behaviours and outcomes.[16 17] Similarities between coaches and patients, coupled with the time spent together, facilitated the establishment of a strong coach–patient relationship.[16] Within this relationship, four key coaching activities emerged: education, personal support, practical assistance and serving as a liaison between patients and clinicians.[16] Furthermore, we have previously explored the experiences of participants from a South Asian background, who were undertaking the National Health Service (NHS) Low Calorie Diet (LCD)

programme.)[18] This study highlighted the importance of adopting a person-centred approach in designing interventions, recognising the influence of culture, motivation and community engagement on health behaviours and outcomes and tailored and culturally appropriate interventions for managing T2D.[18]

However, limited research is available on the experiences of the healthcare service providers in the context of supporting minoritised ethnic participants within T2D management or LCD, total diet replacement (TDR) interventions in the UK.[9 16 17] Furthermore, there is a dearth of evidence relating to the practical challenges, insights and experiences of HCPs or coaches who are working towards an agenda that advocates for culturally relevant support.[19] In this context, in England, the NHS recently piloted a LCD programme to promote remission of T2D.

This paper on coaches' perspectives accompanies our previous paper on participant experiences,[18] building on previous findings from the wider Re:Mission study[20 21] to gain greater understanding of the ethnic inequalities identified.

### The NHS LCD programme

The NHS LCD programme pilot was launched in 2020 (renamed the T2D Path to Remission programme when rolled out nationally in June 2023). Available to adults (18–65 years) living with a body mass index ≥27 kg/m² (adjusted to ≥25 kg/m² for black, Asian and other ethnic groups) and T2D diagnosis within the last 6 years, the NHS LCD programme pilot was initially mobilised in 10 integrated care systems (Integrated care systems are partnerships between NHS bodies, local authorities and local organisations which work together on health and care services to improve the lives of people locally.) (referred to hereon as localities)[22 23] and expanded to a further 11 localities in 2022. The 52-week programme piloted across a total of 21 sociodemographically diverse localities in England, included a 12-week TDR phase which consisted of micronutrient-complete foods such as bars, shakes and soups of various flavours (estimated 800–900 kcals/day) for the first 12 weeks, followed by a 6-week stepped food reintroduction phase and weight maintenance support until programme end[24 25] with behaviour support provided through one of the three models: 1:1, group or digital. Commissioned by NHS England (NHSE), the programme was delivered by six commercial providers, using a predominantly non-medical workforce such as health coaches to deliver the sessions to service users (table 1).

The service specification mandates the promotion of equal access and the tailoring of support through the application of equity-based policy (further description of the programme has been provided elsewhere).[26 27] Preliminary (unpublished) data from the NHS LCD programme highlight that 18% out of an estimated 21% eligible people with Asian ethnicity, 8% out of 8% of those with black ethnicity, vs 66% of those with white ethnicity were referred to the programme.[28] The preliminary data show

**Table 1** Overview of the LCD programme

| Time | Three phases of the LCD programme |
| --- | --- |
| 1–12 weeks | LCD: with total diet replacement products (TDR)c up to 900 calories per day. During this time, patients will replace all normal meals with these products. |
| 12–18 weeks | Food reintroduction: during this time, patients will be supported to reintroduce food, with a stepped phasing out of TDR products. |
| 18–52 weeks | Maintenance, during this time patients will be supported to maintain their initial weight loss. |

LCD, Low Calorie Diet.

that individuals from a South Asian and black ethnicity have lower programme uptake than those of white ethnicity (61% and 67% vs 72%) and lower 12-month weight loss (6.8% vs 10.3%).[28] The overall drop-out rate at 12 months was 55% for those of Asian ethnicity, there were no significant differences by ethnicity in the univariate analyses and in the multivariable logistic regression.[29]

This paper, therefore, aims to explore the experiences of health coaches who deliver the NHS LCD programme to ethnically diverse populations, while identifying the presence or not, of equitable and culturally tailored care. Given the disproportionate impact of T2D on certain ethnic groups, the study bridges the significant knowledge gap regarding the challenges, successes and nuances faced by coaches in culturally tailored diabetes management interventions, in order to inform future service development.

## METHODS

This study formed part of the large Re:Mission study evaluation of the NHS LCD programme.[30] The study received ethical approval from Leeds Beckett University (LBU 106559) and is reported using Consolidated Criteria for reporting Qualitative Research (COREQ) guidelines (see online supplemental file 1). Employing a qualitative research design, this study centres on the subjective views of the coaches while recognising that these experiences will be shaped by underlying structural, cultural and contextual factors.[31] The full research process was conducted over a period of 12 months which included recruitment, interviews, analysis and write-up, with interviews being conducted in March 2023, in order to allow engagement with participants across the three stages of the programme. Data collection and data analysis were carried out by PD who identifies as a female of Indian ethnicity (details on coauthors are reported in online supplemental file 1).

### Materials and procedures

The interview questions broadly covered participants' experience of the programme, such as their training, challenges faced, successful strategies employed and suggestions for programme improvement and cultural tailoring (see online supplemental file 2 for the interview guide). Written informed consent through a completed consent form and sociodemographic information were

obtained prior to the interviews, consent was reconfirmed verbally at the beginning of the interview. All of the semi-structured interviews were conducted by PD via Microsoft Teams, which lasted 30–60 min. All the interviews were audio recorded (with permission); the interviews were transcribed verbatim by an external transcriber. There was no patient and public involvement (PPI) in this study.

### Patient and public involvement

There was no patient and public involvement in this study, however, PPI was involved in the overall Re:Mission project.

### Participants

The recruitment process involved engaging four service providers authorised by the NHSE to deliver the programme. Subsequently, service providers disseminated an email containing the participation information sheet and details about the research to health coaches. These coaches then extended invitations to potential participants, resulting in the purposive recruitment of seven health coaches through two NHSE-authorised service providers. No financial incentive was provided. Ten coaches consented to be interviewed, however, three had to withdraw due to time constraints, and the remaining were all interviewed.

The coaches worked in diverse geographical areas including Frimley, Greater Manchester, Derbyshire, Birmingham and East and North London and delivered the programme to those from ethnic minority backgrounds.[32] One of these recruited participants conducted the programme in a language spoken by South Asian minority groups. Online supplemental file 3 provides background information on the programme and demographic data.

Inclusion criteria included participants delivering the NHS LCD programme either from a minoritised ethnic background or delivering the programme to those from ethnic minority and white British backgrounds. The study proceeded with interviewing all remaining coaches who contacted us and were available to partake in the interview process. Among those recruited, self-reported ethnicity included Pakistani (n=2), Indian (n=1) and white British (n=4). Online supplemental file 3 provides data on how many individuals with T2D were eligible for the programme from different demographic areas in which the pilot programme was based and also the percentage

**Table 2** Participant characteristics

| Coach number | Programme delivery | Programme provider | Self-reported ethnicity | Gender |
|---|---|---|---|---|
| 1 | Online group | 1 | Asian Pakistani | Female |
| 4 | Online group | 2 | Pakistani | Female |
| 6 | Online 1:1 | 2 | British Indian | Male |
| 2 | Online 1:1 | 2 | White British | Female |
| 3 | Online group | 2 | White British | Female |
| 5 | Online 1:1 | 2 | White British | Male |
| 7 | Online group | 2 | White British | Male |

of individuals with T2D from white and minority ethnic backgrounds (table 2).

### Data analysis

Transcripts were transcribed by an external transcriber and checked for accuracy by one researcher (PD). NVivo (V.12) software was used to facilitate data management.

A reflexive thematic analysis, as proposed by Braun and Clarke,[33] was conducted on anonymised interview data used to explore the challenges, successes and personal reflections of coaches in delivering the programme. PD re-read interview transcripts multiple times to gain familiarity with the data while annotating relevant extracts and noting broader ideas that could aid the coding in subsequent stages. Themes were then generated and revised iteratively and discussed with MM who as suggested by Sparkes and Smith acted as a critical friend after each of the stages of analysis to enhance the reflexive self-awareness of interpretations. Field notes were used to enhance this reflective process, at each of the six stages of analysis. Online supplemental file 4 outlines the thematic map.

This methodology enables a flexible process to look at contextually embedded phenomena, such as cultural competence, communication barriers and the impact of coaches' backgrounds on interactions. The study is grounded in a critical realist perspective, which recognises the existence of objective realities while emphasising the importance of centring the subjective experiences and perceptions of underserved populations. This approach acknowledges that while objective structures and conditions shape individuals' lived experiences, these experiences are also influenced by subjective interpretations and perspectives. Further detail on the reflexive process is highlighted in the COREQ checklist (see online supplemental file 1). Malterud et al concept of information power can be used within reflexive thematic analysis as an alternative to data saturation as described by Braun and Clarke.[34 35] This approach allows an interpretive judgement regarding study size related to the purpose and goals of the analysis.

### RESULTS

Five themes were identified, which had a focus on the effectiveness and inclusivity of the programme, these are training and support needs, the impact of language, the use of resources in providing dietary advice, needing to understand the impact of culture and ethnicity and experiences of cultural tailoring.

### Training and support needs

Coaches highlighted the challenges experienced when supporting participants with different ethnicities to their own, in relation to cultural food preferences and considerations associated with their ethnic backgrounds. They discussed limitations in the training they received, including a lack of specific training for working with service users from different ethnic backgrounds and cultures to theirs, and a lack of education about different cultures, multiethnic foods and their impacts on health:

> So I wouldn't say that I had training for ethnic populations. (Asian-Pakistani, female, C1)

> there's not, there's not been much training on the diverse cultures. (White- British, female, C2)

Coaches reported several areas where training and support could be improved, which included a deeper understanding of culturally specific foods, understanding the cultural influences that affect dietary choices, and any additional considerations linked to ethnicity such as festivities and social support. Practical suggestions for training included learning about food groups and common foods within cultures, training days for cultural competencies, education on barriers and influences within specific cultures, and guidance on adapting the programme content to suit participants' cultures.

> the challenges will still be there unless I get some training on foods or how cultures might be affected by the programme. (White-British, female, C2)

They also wanted education and external training from dietitians and health professionals from diverse ethnic backgrounds with an understanding and experience of relevant cultures and lifestyles, to support appreciation and understanding of the nuances within ethnicities. Furthermore, coaches highlighted a desire for support from a workforce that was also ethnically diverse, which would enable peer support between colleagues from different backgrounds to respond to the needs of participants.

I think yeah just having good training is a good thing. I would say training from people of that background. So maybe dieticians who work, you know, are from that ethnicity or that background I think is better… (Asian-Pakistani, female, C1)

## Needing to understand the impact of culture and ethnicity

Coaches described a limited understanding of culture and ethnicity and how these factors could influence the effectiveness of health programmes. They described their knowledge gaps, with one participant reporting uncertainty about what is meant by ethnicity. Furthermore, there was discussion on a lack of awareness of different religions and ethnicities with one participant reporting confusion on what was meant by 'South Asian ethnicity'. Some coaches from a white British background reported difficulty in supporting participants from backgrounds different from their own, due to their limited knowledge of dietary practices and cultural nuances.

> …it is affecting my participants if I'm not fully understanding food, culture, or anything about their background really. (White- British, Female, C2)

> African backgrounds because they tend to have a lot of flour-based foods in their diet and the advice is not really sort of geared to give them practical examples of how they could adjust their diet to make that work. And I, you know I would, I would have to look that up myself, you know, if I was asked that. (White -British, Female, C3)

The gap in their understanding led to some coaches resorting to using a generic approach, with one coach believing that treating every participant the same was better and more ethical, rather than providing tailored advice regarding culture and ethnicity. However, this was regarded at times as being due to the lack of understanding on how to tailor advice and a lack of understanding of the importance of tailoring. Furthermore, some coaches highlighted that before being asked about ethnicity as part of the current study, they had not considered how participants' ethnicity could impact the delivery and methods of the programme.

> …you know I'm still ignorant to a good portion of cultures from around the world. (White- British, male, C7)

> I've never until I was invited for this interview thought about making a distinction between people's backgrounds and their results. So, I'll find that really difficult to answer because I never sort of looked at it in that way… (White- British, Female, C3)

Other coaches acknowledged that a deeper understanding of how food and culture intersect could enhance the quality of the service they provide, with some coaches reporting they consider the difference in cultures when delivering the programme and adapt the delivery according to this.

it is affecting my participants if I'm not fully understanding food, culture, or anything about their background really. (White British, female, C2)

## The impact of language

Coaches from a white British ethnicity described difficulty in communicating with participants due to language differences, creating a challenge in building rapport. It was mentioned that, at times, family members were involved as translators to overcome language barriers, and without this the participants would have faced challenges in understanding the programme's content. In general, coaches' views were that incorporating interpreters and specific language groups within the programme could increase inclusivity and improve adherence. In turn, this would support an environment where individuals with a different preferred language could fully engage and benefit from the programme.

> We need to consider other language groups so that we make it inclusive for everybody. (White British, female, C2)

> …an Urdu specific group to help tailor, to help sort of support those, so I feel like that is something that that, that's good, that helps and can help sort of like reach those that might not be able to join the programme, if because they won't, might not be able to follow it or understand what's going on. (White-British, male, C5)

## The use of resources in providing dietary advice

Experiences of inadequate resources and uncertainty of resources available such as culturally tailored resources were described, particularly from coaches of white British ethnicity. This was described as leading to difficulty in providing culturally tailored guidance. Coaches of both white British and South Asian ethnicity described the resources and information as being centred around a Western diet, which created a barrier when supporting those from other cultural backgrounds.

> a lot of the information is based on a typical Western diet. (White British, female, C3)

> having like maybe different resources so you could go into a conversation with a specific resource open to know what to be talking about in certain areas might be useful. (Pakistani, female, C4)

Coaches highlighted that for resources to be effective, they needed to reflect the cultural background of the programme's participants. They suggested resources can be improved through incorporating culture-specific recipes and further representation in the information and guidance.

> The programme could benefit from a lot more diverse and wide range of recipes (White- British, female, C2)

Maybe making some recipes that would be just tailored towards them. That could be something that we could look at, you know, do. Maybe even and digging into how we can actually break that barrier of you know getting these service users onto the programme without having them to think, oh, these are not the foods that we regularly have so how am I going to have that while I'm sitting with my family? (Pakistani, female, C4)

### Experiences of cultural tailoring

Coaches described a sense of responsibility for adapting programme sessions to align with the cultural backgrounds of participants. Commonly, cultural tailoring was perceived as delivering the content in other languages and providing support and specific resources during Ramadan. Therefore, there was a limited view of cultural tailoring and limited discussions on how different religions may affect the delivery of the NHS LCD programme. Cultural tailoring was at times perceived as providing advice to reduce the quantities of cultural foods such as chapatis.

…it's cutting down on rice and cutting down on chapattis and things. (White-British, male, C7)

Coaches of South Asian ethnicity reported further cultural tailoring through individualised advice and customising the content of the session and also reported the positive impact of sharing the same ethnicity as the participants. These coaches highlighted the benefit of speaking the same language as participants and that their personal experiences and cultural understanding enabled them to relate to participants and build rapport. Furthermore, coaches discussed the importance of flexibility around decision-making as it allowed them to effectively tailor the programme content to the cultural preferences and needs of their participants. An example of this is a coach who reported removing a session about alcohol because it was not relevant to her group of Muslim women (table 3).

…explaining things in a language that other people understood potentially it was quite helpful… (British-Indian, female, C6)

## DISCUSSION

The themes highlight the varying cultural competency of coaches in their understanding of cultures and ethnicities and the potential impact of knowledge gaps such as tailoring on programme delivery. The study provides insight into delivering the LCD programme, emphasising the need for comprehensive training, addressing language barriers, using culturally tailored resources, understanding diverse cultures and implementing effective cultural tailoring strategies. Despite the geographical diversity and two providers involved, consistent themes emerged across locations and among the various service providers interviewed.

## Culturally relevant support

The multifaceted concept of cultural tailoring encompasses linguistic adaptations, cultural understanding and the development of tailored resources, through the use of guiding principles for cultural tailoring and cultural competency to support this.[36] While this research reveals the presence of important 'surface-level' adaptations within the NHS LCD programme such as the Urdu language group and some ethnic-specific resources such as for Ramadan, it also underscores the need for 'deeper' level adaptation, to cultivate a thorough comprehension of the cultural contexts, beliefs and values that influence individuals' health behaviours.[37(p2,3)] 'Deep structure' adaptations are described as those which go beyond superficial adjustments and incorporate understandings of cultural underpinnings and are more likely to result in behaviour change through aligning with frameworks for cultural tailoring such as through creating culturally relevant educational materials and incorporating traditional dietary practices which incorporate understanding of core cultural values and how a range of social, ethnic, environmental and historical beliefs can shape health beliefs and behaviours.[36–38]

The coaches' uncertainty about available resources and the cultural misalignment of existing materials highlight areas where improvements are required. Through enhancing representation, diversifying recipes and providing culturally tailored information and resources, programmes can better meet the needs of ethnically diverse populations. The coaches described the potential benefits of incorporating professional interpreters and establishing specific language-based support groups, ultimately working toward a more inclusive and effective programme for participants from diverse backgrounds. This finding aligns with the theme of the benefit of peer support from the participant paper.[18]

Similarly previous research has highlighted challenges of cultural competency in healthcare and that education, training, interpreter services and diversification of staff would improve their ability to support ethnically diverse individuals.[9] Furthermore, studies exploring the impact of cultural tailoring on diabetes management among ethnic minority populations found that tailored interventions, including language-appropriate resources and culturally sensitive advice, are needed to promote uptake and adherence.[39] The lack of cultural competency in healthcare can exacerbate existing health inequalities within ethnically diverse populations, reflecting a fundamental inequity in access and quality of care.[40] Studies have demonstrated disparities in health outcomes among minority groups, partly due to cultural insensitivity in interventions and healthcare delivery.[41–44]

Addressing this issue necessitates enhancing the cultural competency of healthcare providers, especially coaches involved in delivering health programmes.

**Table 3** Participant quotes and themes

| Quote | Theme | Participant details |
|---|---|---|
| So maybe just having another coach who you can kind of talk to every month whenever you have a session to discuss your group so that they kind of have an insight as well, and someone just to come and watch your group as well. | Training and support needs | C1 (Asian Pakistani, Female) |
| I think even just maybe having some more exposure on the food groups, I think that would be something that we could look into, you know which would help the service users. | Training and support needs | C4 (Pakistani, Female) |
| We took, we've got topics on Ramadan that so we send out like leaflets and information on that, but it's not a topic that I would say I'm fully competent in when you ask me the questions on that. So I don't think we've had enough training, to be honest. | Needing to understand the impact of culture and ethnicity | C2 (White British, Female) |
| I'm very experienced in giving people suggestions, meal suggestions that could be, you know, better choices for diabetes management than the diet that they were having, and not so familiar with, you know, looking for alternatives in, in Asian food. | Needing to understand the impact of culture and ethnicity | C3 (White British, Female) |
| What would be better is having, if people are not fluent in English, having more language, language options on the app or the recipes or the, the use of certain tools in the app. | The impact of language | C2 (White British, Female) |
| providing them with that language, you know, language content, you know, a coach that could speak that language or, you know, somebody that could check in with them even if the coach, if they don't have a coach that's available for their language, just to see how things are. If there is anyone in the office that could, you know speak Urdu, Hindi, Punjabi, what other ethnic, you know Asian ethnic languages are, just to you know see the service user is in comfort when they are joining the session, they don't have any questions, they're not lost. | The impact of language | C4 (Pakistani, Female) |
| Because then they might, they will feel rather than it just all being around maybe focused around sort of like a Western diet, they can then feel like it's more directed for them and they're sort of getting that knowledge that it's going to be more beneficial to them rather than, rather than something that they're not gonna really have much interest in and there only being like, slight, slight add-ons about their own diet. | The use of resources in providing dietary advice | C5 (White British, Male) |
| So we have specific ideas for different cuisines and it wouldn't just be for the South Asian, but it would also have to be for African, Afro Caribbean and those are sort of the main groups that we, yeah that we are aware of. | The use of resources in providing dietary advice | C3 (White British, Female) |
| So you have to convey all of the concepts and ideas and kind of all of that, of course, but it's just how you do it. So, but even things like you know, recipe ideas, we had a lot that were relevant to my group. And also kind of Indian populations as well. | Experiences of cultural tailoring | C1 (Asian Pakistani, Female) |
| I think people do appreciate it when you just put the time in to sort of take on what they're saying and take on, like especially cultural things because people are quite proud of, of their culture a lot of the times. So just, you know, knowing that it's being appreciated, it's being considered, you know, it's not just being brushed off and going OK, well we have this way of doing things so you're gonna abandon your culture basically to eat how we wanna tell you to eat, and that's just not gonna work for anyone. | Experiences of cultural tailoring | C7 (White British, Male) |

Culturally competent interventions have been shown to improve engagement, adherence and outcomes, effectively reducing health disparities.[43 44] By tailoring the NHS LCD programme to be culturally sensitive and equipping coaches with cultural competence, healthcare interventions can become more equitable, ensuring that individuals from diverse ethnic backgrounds receive the same quality of care and just opportunities, as has been demonstrated in previous T2D interventions.[45 46] Theoretically, equity in programme delivery would positively impact outcomes, such as weight loss, promoting fairness and reducing the health disparities prevalent among ethnically diverse populations.[47]

While the coaches did not extensively discuss the format of delivery in interviews, research suggests that despite the convenience and accessibility benefits of online healthcare programmes, ethnically diverse populations may face challenges with this delivery approach.[48] Therefore, providing culturally tailored telehealth services in patients' preferred languages or through bilingual health providers should be considered for more favourable outcomes.[48]

The challenges articulated by the coaches in this research suggest the need for changes at the provider level. Addressing these challenges requires support and training, empowering coaches with the necessary skills and insights to effectively support diverse ethnic populations. Training is required for coaches to enhance their cultural competency to provide the skills to effectively engage with individuals from diverse cultural backgrounds through being able to tailor resources and delivery of the programme to provide culturally relevant

support. This is supported by previous studies which emphasised the significance of cultural competence for HCPs[38 49 50] and reported that culturally tailored dietary advice and resources were associated with increased participant satisfaction and greater adherence to dietary recommendations.[38 50]

### Impact on coach delivery

Coaches described their lack of awareness regarding the impact of cultural competency and tailoring of delivery. Research suggests a variety of healthcare staff particularly in the USA from a white ethnic background may hold unconscious biases towards racial and ethnic minorities[51] and may have preconceptions in regard to a person's background. This can be shaped by their personal experiences and the institutional cultures within healthcare organisations, some of which may have an element of institutional racism.[52] There is a need for enhancing cultural competence which goes beyond typical judgements of ethnicities or cultures. Rather, practitioners need an understanding not only of their own perceptions and attributions of others but also a critical self-reflection of how these perceptions are created.[52 53] This would support coaches in exploring how factors impact patients whose values and priorities may differ from their own.[53 54]

Therefore, the pressing need for training becomes evident, as highlighted by coaches who have identified a lack of self-knowledge concerning cultural nuances and dietary preferences, yet find they need to tailor the programme to meet the needs of the diverse population group without relevant skills and knowledge of cultural differences. The need for cultural tailoring in healthcare interventions is underscored by previous research with diverse ethnic participants undertaking the NHS LCD, which emphasised the necessity for individualised advice and tailored support to address the unique needs of diverse populations, ensuring that healthcare services are effective and equitable for all individuals.[18] The study highlighted the important and complex role of the coach–participant relationship while many praised the supportive role of coaches, acknowledging their empathy, effective communication skills and provision of resources, others encountered challenges stemming from a lack of cultural understanding.[18]

Research shows that health coaches' attitudes, communication styles and ability to understand and implement cultural nuances impact patient–provider relationships in healthcare programmes.[16] There is a need to support individuals with a person-centred approach while considering the wider determinants of health and communicating with each individual to find out the best way to support them to provide equitable care. However, as shown from the results in this paper there are difficulties in communication at times, especially where English is not the first language.

South Asian coaches in this study described the benefit of cultural matching in which HCP and participant are matched based on ethnicity. While cultural matching between HCPs and participants has previously been described as helpful to allow for shared understanding, it may not always be feasible given the lack of funding in healthcare, the diverse range of cultural backgrounds among participants and the need for competent and experienced HCPs.[55] Therefore, focusing on cultural competency through training and resource provision for those delivering the programme would support in providing tailored advice which is equitable. Such training can equip HCPs with the skills and insights necessary to give culturally sensitive advice to support those from diverse ethnic backgrounds. However, it is important to note an exclusive focus on culture or ethnicity can lead potentially to assumed judgements and stereotypes. This was highlighted by coaches in the lack of understanding of ethnicities, cultural differences and the need for cultural considerations in health programmes on the basis of ethnicity.[56] Instead, a person-centred approach which considers social, environmental and contextual factors is required.[10]

### Strengths and limitations

To our knowledge, this is the first study of this kind, providing insights into coaches' experience on an LCD programme an area of research which is underrepresented within T2D.

A limitation of this study is that despite intending to do so, there was a lack of uptake and thus a lack of representation of service providers with only two out of six being represented. A further limitation includes the collective analysis of coaches, overlooking their diverse characteristics and varying backgrounds, which could impact their experiences of delivery as there are differences in heritage, cultural practices, generational and regional differences. However, not all background information could not be collected to maintain the anonymity of participants.

Additionally, the duration of coaches' experience in delivering the programme was unspecified and there is a lack of detail regarding the ratio of coaches to the total patient ethnicity or the percentage of different ethnicities managed by each coach which could impact on their experiences, therefore, collecting this information in future research would be useful. Another limitation is the lack of background information available about the coaches and service providers, raising questions about the representativeness of the sample in relation to the broader target population. Furthermore, a limitation of this research study is the insufficient exploration of the substantial variations in ethnicities and cultures, the focus was on the coaches, however, is it a limitation of the research that it was not possible to consider the nuanced, diverse needs within and between ethnic and cultural groups. The absence of in-depth discussions or data collection regarding these diverse subcategories highlights the need for further research and considerations

to ensure a comprehensive understanding of how cultural nuances may impact the implementation and effectiveness of the LCD.

## Implications for policy and practice

The findings from this study have implications for NHS LCD programme delivery and future policy and practice within the healthcare sector, particularly in the context of supporting service providers in providing culturally tailored and effective approaches to diverse ethnic populations. To support more equitable healthcare interventions, the following actions are recommended:

► Cultural competency training: Incorporate comprehensive cultural competency training into the education and professional development of coaches, supporting better understanding of cultural nuances, dietary preferences and the influence of culture on health behaviours. Coaches need to be able to deliver the programme to those from different backgrounds and ethnicities, to effectively do this an understanding of cultural nuances is required.

► Culturally tailored resources: Development and utilisation of culturally tailored resources and materials for healthcare interventions, aligning with the dietary preferences and cultural backgrounds of participants. For example, signposting to existing resources and literature such as the South Asian and African and Caribbean Eatwell guide.[57]

► Language-specific groups: Develop language-specific support groups and resources within healthcare programmes to cater to participants where English is not a first language.

► Diverse workforce: Within the contextual setting, the programme is being delivered to promote diversity within the workforce. Having colleagues from diverse backgrounds facilitates mutual support, encouraging the exchange of knowledge and fostering interpersonal collaboration. This diversity can enable team members to ask questions, share insights and collectively enhance their cultural competence, ultimately ensuring more effective support for the ethnically diverse participant population.

► Further research: Is needed to examine cultural and contextual influences, the long-term effects of cultural tailoring on patient outcomes, the most effective methods for cultural competency training and the scalability of culturally sensitive interventions within broader healthcare systems. Through exploring this in future research, programmes and practices within healthcare can be improved and a more equitable service provided.

► Future research: Should detail the ratio of coaches to the total patient ethnicity or reflect the percentage of different ethnicities managed by each coach, which would provide valuable insights for a more nuanced understanding of programme effectiveness and impact across diverse populations.

## Conclusion

In healthcare programmes, care should be person-centred, with lifestyle interventions involving culturally tailored approaches, particularly in relation to dietary and lifestyle advice. There is a need to develop an understanding of different cultures and ethnicities and the factors that may contribute to inequity for different populations to support the delivery of future health programmes. By acknowledging the current limitations in training, these findings advocate for a more culturally competent approach to the education and support of health coaches, to foster more inclusive healthcare practices. Among other recommendations including culturally tailored resources and language-specific groups, coaches delivering the NHS LCD programme would benefit from enhanced training and support to equip them with the knowledge and skills necessary to tailor delivery of the programme to individuals from diverse ethnic backgrounds: reducing barriers and optimising inclusivity and effectiveness of the programme across ethnic groups.

**Contributors** PD, MM, CVH, KJD and LJE provided substantial contributions to the design of the work. PD, MM and KJD provided substantial contributions to the interpretation of data for the work. All authors PD, MM, KJD, CVH, LJE and CB drafted the work and provided revisions. All authors provided final approval of the version to be published and provided agreement to be accountable for all aspects of the work in ensuring that questions related to the accuracy or integrity of any part of the work are appropriately investigated and resolved. PD is the guarantor of this work.

**Funding** This work was supported by the National Institute for Health Research, Health Services and Delivery Research (NIHR 132075). The NHS England Low Calorie Diet Pilot is funded by NHS England.

**Competing interests** All authors confirm that they have no conflicts of interest to declare. LJE has received funding from NIHR, MRC, Leeds Council and OHID/PHE in the last 3 years and has had an honorary contract with OHID. CB is a primary care advisor to the national diabetes programme for NHS England.

**Patient and public involvement** Patients and/or the public were not involved in the design, or conduct, or reporting, or dissemination plans of this research.

**Patient consent for publication** Not applicable.

**Ethics approval** This study involves human participants and ethical approval was gained from Leeds Beckett University (LBU 106559), written informed consent was obtained from participants. The Re:Mission study was approved by Health Research Authority on 5 July 2021, REC ref: 21/WM/0136. Participants gave informed consent to participate in the study before taking part.

**Provenance and peer review** Not commissioned; externally peer reviewed.

**Data availability statement** Data are available on reasonable request. The datasets generated during this current study are not publicly available due to reasons of privacy and confidentiality, and because of the inability to deidentify the data. Additional knowledge of the data can be available from the corresponding author on reasonable request.

**ORCID iDs**
Pooja Dhir http://orcid.org/0000-0001-9225-0442
Maria Maynard http://orcid.org/0000-0002-0011-752X

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
