## [Reviewer comments · BMJ Open]

ARTICLE DETAILS

TITLE (PROVISIONAL)	A qualitative evaluation in community settings in England exploring the experiences of coaches delivering the NHS Low Calorie Diet Programme Pilot to ethnically diverse participants.
AUTHORS	Dhir, Pooja; Maynard, Maria; Drew, Kevin J; Homer, Catherine; Bakhai, Chirag; Ells, Louisa

VERSION 1 – REVIEW

REVIEWER	Barzilay, Joshua Kaiser Permanente, Endocrinology
REVIEW RETURNED	17-Feb-2024

GENERAL COMMENTS	This paper summarizes the experiences of 6 people delivering health advice to a mixed population of people of different backgrounds. While the paper is fine, please shorten it. The paper is much, much too long and repeats itself. You can say what you want to say in less than half the space that is currently used. Be to the point
---

REVIEWER	Azami, Golnaz University Putra Malaysia, Nursign and Rehabilitation
REVIEW RETURNED	21-Feb-2024

GENERAL COMMENTS	Subject: Review Feedback - "Supporting ethnically diverse populations within a national diabetes intervention; a qualitative evaluation of the experiences of coaches delivering the NHS Low Calorie Diet Programme Pilot" Dear Authors, Thank you for the opportunity to review your manuscript, "Supporting ethnically diverse populations within a national diabetes intervention; a qualitative evaluation of the experiences of coaches delivering the NHS Low Calorie Diet Program Pilot." I found the topic to be relevant and appreciate your efforts in contributing to this field. While the overall manuscript is well-written and informative, I have some suggestions for improvement that I believe would strengthen your work. Please consider the following: Title of the manuscript:
--

	To ensure clarity for a broader audience, I suggest avoiding abbreviations unless they are universally understood within the field. Utilizing full terms initially, followed by the abbreviation in parentheses, can enhance readability and comprehension Methodology section What methodological orientation was stated to underpin the study? e.g. grounded theory, discourse analysis, ethnography, phenomenology, content analysis How many people refused to participate or dropped out? Reasons? Keep up the good work!
--	---

REVIEWER	Babiker, Rasha National University, Physiology Department
REVIEW RETURNED	29-Feb-2024

GENERAL COMMENTS	This paper is well-written and significant in shaping the lifestyle of diabetic patients. The author evaluated the coaches' experience in delivering the NHS Low Calories Diet Programme to a diverse population. They recommended using a larger sample size in future studies was clearly stated. Furthermore, it would be beneficial if the author could provide a broad explanation of the required sample size, including the inclusion of different ethnicities and the minimum duration of experience necessary for effective programme delivery. Additionally, the title of Table 2 should be removed for clarity (line 161 to line 152). Although the methodology is clearly described, the participation of only seven coaches represents a small sample size. Future research could benefit from detailing the ratio of coaches to the total patient ethnicity or reflecting the percentage of different ethnicities managed by each coach.
--

REVIEWER	Piya, Milan Western Sydney University
REVIEW RETURNED	11-Mar-2024

GENERAL COMMENTS	Dhir and colleagues have conducted qualitative interviews with seven health coaches delivering the NHS LCD programme and highlighted the need for more training and support for health coaches to deliver a culturally tailored intervention. The findings are interesting and informative but comments are included below, some of which are acknowledged by the authors in the limitations section of the paper.  1. While the title does suggest that this was an evaluation of the experiences of health coaches, there is a not much mention of the participant experience in this matter, and perhaps more reference to the views of participants would be helpful. There is a separate piece of work that has just been mentioned in passing by the authors and would be helpful to link that work to this in the introduction and discussion. 2. The sample size of seven interviews is not great, and while that may not be a problem in itself, the heterogeneity in the participants (sex and ethnicity), perhaps more interviews would have helped improve the data saturation and capture more themes? Also, as the authors mention, not all service providers of the NHS LCD were represented so how generalisable are these findings?
---

	3. It looks like all of the health coaches provided online delivery of the programme, which in itself may be an issue for an ethnically diverse population. There is no mention of this in the discussion and it would be good to know what the authors feel about that. Minor comments  1. In the introduction first paragraph, why is the prevalence of T2DM a public health concern only in Western and industrialised societies? Isn't it a global problem? ANd why is food only culturally significant in minority ethnic groups? And there is no mention of language (spoken and written for support material) as a factor in this paragraph or later in the introduction, when it is clearly an important component (and also confirmed in the results). 2. The NHS LCD program was underpinned by the DIRECT study. Please see participant experience in Rehackova et al PMID: 34519099, which may inform some of this research although it was conducted in the general population. Also, the qualitative research of participants and health care professionals may inform some of the research as well done from DiRECT Australia by Chimoriya et al PMID: 38311881 as these were both conducted in similar contexts of low calorie diet followed by food reintroduction. 3. While the sample size is quite small, were the authors able to identify if the themes were similar across the different locations/sites? 4. Was there any mention of religion e.g. in South Asian populations? And how this may affect the delivery of the NHS LCD program? This would probably affect the resources and types of food suggested? 5. It seems from these interviews that health coaches from an ethnically diverse background seemed better equipped to deliver a culturally sensitive program? What do the authors feel about the use of a diverse workforce in terms of ethnically diverse health coaches. 6. What do the authors feel about peer learning within these groups and grouping people of similar ethnicity together where possible to make the language, information and food types to be modified to suit them?
--	---

VERSION 1 – AUTHOR RESPONSE

Reviewer 1:

This paper summarizes the experiences of 6 people delivering health advice to a mixed population of people of different backgrounds.

While the paper is fine, please shorten it. The paper is much, much too long and repeats itself. You can say what you want to say in less than half the space that is currently used. Be to the point.

Thank you for your feedback. We have carefully reviewed the paper and incorporated amendments based on the helpful comments and feedback from all the reviewers. Our revisions focus on eliminating repetition and presenting concise points while ensuring the qualitative nature of the study is preserved.

Reviewer 2:

Title of the manuscript:

1. To ensure clarity for a broader audience, I suggest avoiding abbreviations unless they are universally understood within the field. Utilizing full terms initially, followed by the abbreviation in

parentheses, can enhance readability and comprehension

Thank you for this helpful feedback. The title has been changed to 'A qualitative evaluation in community settings in England exploring the experiences of coaches delivering the National Health Service Low Calorie Diet Programme Pilot to ethnically diverse participants.'

Methodology section

2. What methodological orientation was stated to underpin the study? e.g. grounded theory, discourse analysis, ethnography, phenomenology, content analysis

Thank you for highlighting this, the following methodological orientation has now been added to page 11, lines 266-271 'The study is grounded in a critical realist perspective, which recognises the existence of objective realities while emphasising the importance of centering the subjective experiences and perceptions of underserved populations. This approach acknowledges that while objective structures and conditions shape individuals' lived experiences, these experiences are also influenced by subjective interpretations and perspectives.'

3. How many people refused to participate or dropped out? Reasons?

This is currently described on page 9, lines 234-236 'Ten coaches consented to be interviewed however three had to withdraw due to time constraints, the remaining were all interviewed.'

Reviewer 3:

They recommended using a larger sample size in future studies was clearly stated. Furthermore, it would be beneficial if the author could provide a broad explanation of the required sample size, including the inclusion of different ethnicities and the minimum duration of experience necessary for effective programme delivery.

In the context of this qualitative research study, the sampling methodology employed is purposive, targeting individuals with lived experiences as coaches either of a diverse ethnic background or delivering the programme to ethnically diverse individuals. Therefore there was no minimum duration of experience required or a required sample size, as the study sought to attain qualitative data providing depth of insight into the experiences and perspectives of the coaches, rather than quantitative data for statistical analysis. By adopting this approach, the research aims to capture the rich and nuanced aspects of the coaches roles within the programme delivery context. This is reflected on page 3, lines 68-70 'A potential limitation lies in the small sample size of seven participants; however, it is important to recognise that in qualitative research the emphasis is placed on the depth and richness of the data rather than on statistical power.

This is further substantiated, on page 11 and lines 272-275 'Malterud, Siersma, and Guassora's (2016) concept of information power can be used within reflexive thematic analysis as an alternative to data saturation as described by Braun and Clarke (2021).^{1,2} This approach allows an interpretive judgement regarding study size related to the purpose and goals of the analysis.

Additionally, the title of Table 2 should be removed for clarity (line 161 to line 152).

This table has now been reformatted for clarity.

Although the methodology is clearly described, the participation of only seven coaches represents a small sample size. Future research could benefit from detailing the ratio of coaches to the total patient ethnicity or reflecting the percentage of different ethnicities managed by each coach.

Thank you for this feedback, this is a very helpful suggestion. Whilst we acknowledge that additional background information from service providers and coaches would be beneficial, unfortunately such data was not available for this study. However, it is noteworthy for future research.

The following has been added to the limitations on page 21 and lines 528-532 'Additionally, the duration of coaches' experience in delivering the programme was unspecified and there is a lack of detail regarding the ratio of coaches to the total patient ethnicity or the percentage of different ethnicities managed by each coach which could impact on their experiences, therefore collecting this information in future research would be useful.'

Furthermore, this limitation has been added as a future research recommendation on page 23 and lines 575-578 'Future research: Should detail the ratio of coaches to the total patient ethnicity or reflect the percentage of different ethnicities managed by each coach, which would provide valuable insights for a more nuanced understanding of programme effectiveness and impact across diverse populations.'

Reviewer 4:

1. While the title does suggest that this was an evaluation of the experiences of health coaches, there is a not much mention of the participant experience in this matter, and perhaps more reference to the views of participants would be helpful. There is a separate piece of work that has just been mentioned in passing by the authors and would be helpful to link that work to this in the introduction and discussion.

Thank you for this helpful suggestion which has been addressed in the following changes to the introduction and discussion, which we hope now ties this paper in with the previous paper in which ethnically diverse participants were interviewed:

Introduction page 6 and lines 150-155 the following has been added 'Furthermore, we have previously explored the experiences of participants from a South Asian background, who were undertaking the NHS Low Calorie Diet (LCD) programme. This study highlighted the importance of: adopting a person-centred approach in designing interventions, recognising the influence of culture, motivation, and community engagement on health behaviours and outcomes, and tailored and culturally appropriate interventions for managing T2D.3

Discussion: the following has been added to page 20 and lines 492-495: 'The study highlighted the important and complex role of the coach-participant relationship: while many praised the supportive role of coaches, acknowledging their empathy, effective communication skills, and provision of resources, others encountered challenges stemming from a lack of cultural understanding.'

2. The sample size of seven interviews is not great, and while that may not be a problem in itself, the heterogeneity in the participants (sex and ethnicity), perhaps more interviews would have helped improve the data saturation and capture more themes? Also, as the authors mention, not all service providers of the NHS LCD were represented so how generalisable are these findings?

This comment has been addressed in our response to reviewer 3 above. The lack of uptake of interviewees and the heterogeneity of coaches is also currently highlighted on page 21 and line 521-526 'A limitation of this study is that despite intending to do so, there was a lack of uptake and thus a lack of representation of service providers with only two out of six being represented. A further limitation includes the collective analysis of coaches, overlooking their diverse characteristics and varying backgrounds, which could impact on their experiences of delivery as there are differences in heritage, cultural practices, generational and regional differences. However, not all background information could not be collected to maintain the anonymity of participants.'

3. It looks like all of the health coaches provided online delivery of the programme, which in itself may be an issue for an ethnically diverse population. There is no mention of this in the discussion and it

would be good to know what the authors feel about that.

We appreciate this insight and have added the following to the discussion on page 19, line 456-461 to reflect this 'Whilst the coaches did not extensively discuss the format of delivery in interviews , research suggests that despite the convenience and accessibility benefits of online healthcare programmes, ethnically diverse populations may face challenges with this delivery approach.⁴ Therefore providing culturally tailored telehealth services in patients' preferred languages or through bilingual health providers should be considered for more favourable outcomes.⁴

Minor comments

1. In the introduction first paragraph, why is the prevalence of T2DM a public health concern only in Western and industrialised societies? Isn't it a global problem? ANd why is food only culturally significant in minority ethnic groups? And there is no mention of language (spoken and written for support material) as a factor in this paragraph or later in the introduction, when it is clearly an important component (and also confirmed in the results).

The following has been changed in the introduction on page 4 and lines 101-105 to reflect this helpful observation 'The prevalence of type 2 diabetes (T2D) remains a significant global public health concern^{5,6}, and its management requires a multifaceted lifestyle approach that encompasses dietary modifications.⁵⁻⁷ However, food is culturally significant, where it often serves as a cornerstone of traditions, identity, and social cohesion.⁷

And on page 5, lines 131-139 has been added 'Research underscores the significance of tailoring health programme advice not only in terms of language but also in culturally appropriate contexts, encompassing written support materials and verbal interactions.⁸ Studies emphasise the role of culturally competent communication in healthcare settings, highlighting its positive impact on patient outcomes and satisfaction and indicate that culturally tailored interventions lead to improved health outcomes among diverse populations.^{8,9} Therefore, incorporating culturally appropriate language and materials into health programmes is essential for promoting effective engagement and understanding, which can ultimately contribute to better health outcomes and patient experiences.⁹'

2. The NHS LCD program was underpinned by the DIRECT study. Please see participant experience in Rehackova et al PMID: 34519099, which may inform some of this research although it was conducted in the general population. Also, the qualitative research of participants and health care professionals may inform some of the research as well done from DiRECT Australia by Chimoriya et al PMID: 38311881 as these were both conducted in similar contexts of low calorie diet followed by food reintroduction.

This study builds upon the broader findings of the RE:MISSION study, which involved interviewing various groups about their experiences working within the programme, this research identified barriers such as ethnicity, culture, language, translation needs, digital competency, and family requirements. As a result of these broad findings this paper specifically explores the perspectives of coaches to gain additional insights, that complement our participant-focused paper, and further exploring inequalities arising from the programme.

The following has been added on page 6, lines 163-165 'This paper on coaches' perspectives accompanies our previous paper on participant experiences³, building on previous findings from the wider re:mission study^{10,11}, to gain greater understanding of the ethnic inequalities identified.

3. While the sample size is quite small, were the authors able to identify if the themes were similar across the different locations/sites?

Thank you for highlighting this. Service provision occurs across multiple sites with various providers,

rather than adhering to a one-provider, one-site model. Interviews conducted with these providers encompassed a diverse range of geographic locations. The following has been added to page 17 lines 412-414 'Despite the geographical diversity and two providers involved, consistent themes emerged across locations and among the various service providers interviewed.'

4. Was there any mention of religion e.g. in South Asian populations? And how this may affect the delivery of the NHS LCD program? This would probably affect the resources and types of food suggested?

The only mention of religion was in relation to the religious holiday of Ramadan, this is reflected in the results on pages 14-15, lines 385-388 'Commonly, cultural tailoring was perceived as delivering the content in other languages and providing support and specific resources during Ramadan. Therefore, there was a limited view of cultural tailoring and limited discussions on how different religions may affect the delivery of the NHS LCD programme.'

Findings relating to tailoring of resources and food types is already captured in:

Theme 1 on page 11 and lines 290-293 'Coaches reported several areas where training and support could be improved, which included a deeper understanding of culturally specific foods, understanding the cultural influences that affect dietary choices, and any additional considerations linked to ethnicity such as festivities and social support.'

And theme 2 on page 13 and lines 337-340: 'Other coaches acknowledged that a deeper understanding of how food and culture intersect could enhance the quality of the service they provide, with some coaches reporting they consider the difference in cultures when delivering the programme and adapt the delivery according to this.'

And theme 5 on pages 14-15 and lines 385-390 'Commonly, cultural tailoring was perceived as delivering the content in other languages and providing support and specific resources during Ramadan. Therefore, there was a limited view of cultural tailoring and limited discussions on how different religions may affect the delivery of the NHS LCD programme. Cultural tailoring was at times perceived as providing advice to reduce the quantities of cultural foods such as chapatis.'

5. It seems from these interviews that health coaches from an ethnically diverse background seemed better equipped to deliver a culturally sensitive program? What do the authors feel about the use of a diverse workforce in terms of ethnically diverse health coaches.

We agree that coaches from an ethnically diverse background were able to provide further cultural tailoring and support participants in their first languages. This is highlighted on page 12 and lines 315-317 in theme 2: 'Some coaches from a White British background reported difficulty in supporting participants from backgrounds different from their own, due to their limited knowledge of dietary practices and cultural nuances.'

And theme 5 on page 15 and lines 393-400 'Coaches of South Asian ethnicity reported further cultural tailoring through individualised advice and customising the content of the session, and also reported the positive impact of sharing the same ethnicity as the participants. These coaches highlighted the benefit of speaking the same language as participants and that their personal experiences and cultural understanding enabled them to relate to participants and build rapport. Furthermore, coaches discussed the importance of flexibility around decision-making as it allowed them to effectively tailor the programme content to the cultural preferences and needs of their participants.'

And theme 1 on page 11 and lines 302-304 'Furthermore, coaches highlighted a desire for support from a workforce that was also ethnically diverse, which would enable peer support between colleagues from different backgrounds to respond to the needs of participants.'

The benefits of ethnic matching is also discussed on page 20-21 and lines 503-510- 'South Asian coaches in this study described the benefit of cultural matching in which HCP and participant are matched based on ethnicity. Whilst cultural matching between HCPs and participants has previously been described as helpful to allow for shared understanding, it may not always be feasible given lack

of funding in healthcare, the diverse range of cultural backgrounds amongst participants and the need for competent and experienced HCPs.¹² Therefore focusing on cultural competency through training and resource provision for those delivering the programme would support in providing tailored advice which is equitable.

6. What do the authors feel about peer learning within these groups and grouping people of similar ethnicity together where possible to make the language, information and food types to be modified to suit them?

We agree that peer support is highly beneficial as it was a key theme in the participant paper with South Asian individuals undertaking the NHS LCD programme, and Highlighted in this paper on page 18 and lines 431-435- 'The coaches described the potential benefits of incorporating professional interpreters and establishing specific language-based support groups, ultimately working toward a more inclusive and effective programme for participants from diverse backgrounds. This finding aligns with the theme on the benefit of peer support from the participant paper.³

There was some mention of peer support groups within the NHS programme from some coaches perspectives in that it may be helpful for participants however, it was not extensively discussed as reflected in the lack of discussion in the paper.

1. Malterud K, Siersma VD, Guassora AD. Sample Size in Qualitative Interview Studies: Guided by Information Power. *Qual Health Res.* Nov 2016;26(13):1753-1760. doi:10.1177/1049732315617444
2. Braun V, Clarke V. To saturate or not to saturate? Questioning data saturation as a useful concept for thematic analysis and sample-size rationales. *Qualitative Research in Sport, Exercise and Health.* 2021/03/04 2021;13(2):201-216. doi:10.1080/2159676X.2019.1704846
3. Dhir P, Maynard M, Drew KJ, Homer CV, Bakhai C, Ells LJ. South Asian individuals' experiences on the NHS low-calorie diet programme: a qualitative study in community settings in England. *BMJ Open.* 2023;13(12):e079939. doi:10.1136/bmjopen-2023-079939
4. Truong M, Yeganeh L, Cook O, Crawford K, Wong P, Allen J. Using telehealth consultations for healthcare provision to patients from non-Indigenous racial/ethnic minorities: a systematic review. *Journal of the American Medical Informatics Association.* 2022;29(5):970-982. doi:10.1093/jamia/ocac015
5. NHS. Long Term Plan. Updated 19/09/2023. Accessed 03rd July, 2023. <https://www.longtermplan.nhs.uk/>
6. Organisation WH. Obesity and overweight. World Health Organisation. Accessed 10th December, 2022. <https://www.who.int/news-room/fact-sheets/detail/obesity-and-overweight>
7. Leung G, Stanner S. Diets of minority ethnic groups in the UK: influence on chronic disease risk and implications for prevention. *Nutrition Bulletin.* 2011;36(2):161-198. doi:10.1111/j.1467-3010.2011.01889.x
8. Saha S, Beach MC, Cooper LA. Patient Centeredness, Cultural Competence and Healthcare Quality. *Journal of the National Medical Association.* 2008;100(11):1275-1285. doi:10.1016/s0027-9684(15)31505-4
9. Harrison R, Walton M, Chauhan A, et al. What is the role of cultural competence in ethnic minority consumer engagement? An analysis in community healthcare. *International Journal for Equity in Health.* 2019;18(1)doi:10.1186/s12939-019-1104-1
10. Jones S, Brown TJ, Watson P, et al. Commercial provider staff experiences of the NHS low calorie diet programme pilot: a qualitative exploration of key barriers and facilitators. *BMC Health Services Research.* 2024;24(1)doi:10.1186/s12913-023-10501-y
11. Homer C KK, Drew KJ, et al. A qualitative evaluation of experiences of the Total Diet Replacement phase of the NHS Low Calorie Diet Programme pilot. *British Journal of Diabetes.* 2024;24doi:10.15277/bjd.2024.435
12. Handtke O, Schilgen B, Mösko M. Culturally competent healthcare - A scoping review of

strategies implemented in healthcare organizations and a model of culturally competent healthcare provision. PLoS One. 2019;14(7):e0219971. doi:10.1371/journal.pone.0219971

VERSION 2 – REVIEW

REVIEWER	Azami, Golnaz University Putra Malaysia, Nursign and Rehabilitation
REVIEW RETURNED	26-Apr-2024

GENERAL COMMENTS	Dear authors, Hello, thank you for submitting your manuscript “A qualitative evaluation in community settings in England exploring of the experiences of coaches delivering the NHS Low Calorie Diet Programme Pilot to ethnically diverse participants” to BMJ Open. After carefully assessing the revised manuscript, I feel that it has potential for publication in its current format. Congratulations and keep up the good work!
---

REVIEWER	Piya, Milan Western Sydney University
REVIEW RETURNED	15-Apr-2024

GENERAL COMMENTS	Thank you for addressing the comments of all the reviewers.
---

VERSION 2 – AUTHOR RESPONSE

Please include a methodological strength in your Strength and Limitations section

- The following has been added to this section 'A strength of this study is its utilisation of one-to-one semi-structured interviews, allowing for an in-depth exploration of the experiences of health coaches delivering the NHS LCD programme to individuals from diverse ethnic backgrounds.'

Please revise point three of your Strength and Limitations section as it focuses on the studies impact

- This has been deleted from the paper.

Please remove the "What is already known about this subject" as it is not part of the journal formatting

- This has been deleted from the paper.